# DXFeat: Depth-Aware Features for Robust Image Matching

## Abstract

This study introduces DXFeat, a novel architecture that integrates depth information as an auxiliary branch for keypoint detection, leveraging depth cues to enhance localization accuracy, which improves localization accuracy with an average 3.1% gain while preserving inference efficiency. DXFeat refines feature extraction during training while maintaining computational efficiency. The model incorporates a modified reliability loss and learnable weighting mechanisms, balancing accuracy and robustness. By optimizing network channels while preserving high-resolution inputs, DXFeat supports both sparse and semi-dense matching, making it well-suited for visual localization and augmented reality. A depth-assisted refinement module further enhances feature representation using coarse local descriptors. Notably, the depth auxiliary branch is only needed during training, ensuring streamlined deployment. Comprehensive evaluations on MegaDepth, ScanNet, and HPatches confirm that the combination of loss-level optimization and depth-auxiliary refinement yields consistent AUC improvements, establishing DXFeat as a strong and efficient framework for real-world image matching tasks.

## 1 Introduction

In high-level computer vision applications, image feature extraction is not just fundamental, but absolutely critical to success. Despite the remarkable advancements that deep learning has brought to the field, especially in tackling the challenges of image matchingEdstedt et al. (2024b); Wang et al. (2024); Sun et al. (2021), many of these methods demand significant computational resources. This presents a substantial roadblock for real-world applications, particularly in fields like roboticsMur-Artal & Tardós (2017), autonomous navigation, and embedded systems, which continue to face major difficulties in efficiently utilizing these resource-intensive approaches. While recent research has focused on optimizing network architectures to address these limitations, there is still a vast untapped potential for further enhancement, particularly in the quality of feature extraction and local descriptors. The existing methods often fall short in balancing performance with efficiency, creating a significant gap that needs to be addressed for practical deployment in resource-constrained environments Inspired by the latest breakthroughs in image matching and depth estimation, we address a critical challenge in feature extraction: The inconsistencies in focal length and depth resulting from camera movement, which frequently lead to keypoint loss. To overcome this challenge, we introduce DXFeat, a lightweight yet highly effective architecture built upon the foundation of XFeatPotje et al. (2024), with the addition of a depth-assisted branch. By incorporating depth information into the feature extraction process, our approach significantly enhances keypoint detection and local feature extraction, improving overall performance while ensuring that it remains computationally efficient on existing hardware.

Keypoint-based methods are particularly advantageous for efficient visual localization when using Structure-from-Motion (SfM) mapsSarlin et al. (2019), while dense feature matchingEdstedt et al. (2023a) excels in estimating camera poses in textureless environments. By combining these strengths, DXFeat effectively bridges the gap between the two paradigms.

In comparison to current image correspondence methods, DXFeat delivers a marked improvement in matching accuracy. It outperforms existing lightweight, deep learning-based local feature alternatives by approximately 3% in precision and achieves performance on par with, or even surpassing,

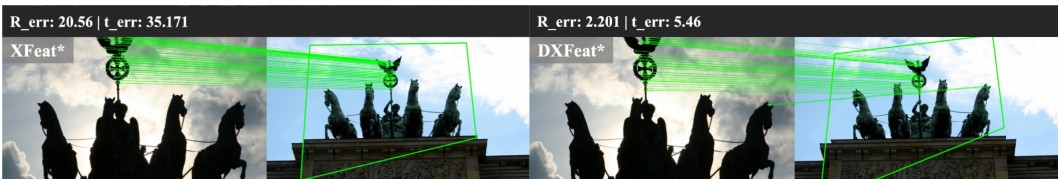

Figure 1: The left image demonstrates keypoint matching under different viewpoint variations. The right image shows the depth map output of DXFeat. This method generates depth information to assist image matching, thereby improving robustness in the matching process.

Figure 2: Compares XFeat*(left) with DXFeat*(right). We observed that DXFeat* provides more reliable keypoints compared to XFeat*, resulting in significantly smaller rotational and translational errors. The green square grid represents the camera viewpoint projected back after the perspective transformation, highlighting DXFeat*'s superior accuracy and robustness in feature matching.

state-of-the-art models like SiLKGleize et al. (2023) and ALIKEZhao et al. (2022) in terms of accuracy. To further elevate precision without sacrificing competitive accuracy, our work introduces two pivotal contributions:

- We introduce a depth-assisted keypoint detection branch that enhances keypoint detection and local descriptors. This efficient branch, removable during downstream tasks, achieves speed similar to XFeat while improving accuracy. Validated across multiple datasets, it shows effectiveness in visual localization, camera pose estimation, and homography construction.
- We modify the reliability loss function and introduce learnable weights into the XFeat representation, enabling the network to assess the importance of different levels during distillation-based matching. These changes improve local feature descriptors and enhance the model's generalizability.

## 2 RELATED WORK

**Early Work in Keypoint Detection and Matching:** Early approaches relied on hand-crafted methods such as Harris corners (Harris et al., 1988), SIFT (Lowe, 2004), and ORB (Rublee et al., 2011), which exploited geometric cues like corners and scale-space extrema. These techniques remain efficient and competitive today. With the rise of learning-based methods, SuperPoint (DeTone et al., 2018) introduced synthetic data for training but required complex procedures. SiLK (Gleize et al., 2023) advanced this direction with an end-to-end probabilistic framework based on double-softmax cycle consistency, achieving strong accuracy without relying on context aggregation (CA) modules such as those in SuperGlue (Sarlin et al., 2020), though CA integration may further improve performance. Other methods have introduced specialized improvements: DeDoDe (Edstedt et al., 2023b) and its successor DeDoDev2 (Edstedt et al., 2024a) add dual-domain processing and geometric constraints for robustness; Darkfeat (He et al., 2023) enhances descriptors in low-light scenes; and R2D2 (Revaud et al., 2019) improves repeatability and reliability via unsupervised learning. Beyond 2D, NeRF-based methods (Youssef & Vasconcelos, 2024) leverage neural radiance fields for high-quality 3D reconstruction and keypoint alignment. Collectively, these advances highlight the shift toward learning-based, task-adaptive methods that continue to push the accuracy and robustness of keypoint detection and matching.

**Image Matching and Computational Efficiency** Modern image matching methods combine traditional keypoint detection with deep learning for local patch descriptors or joint keypoint detection

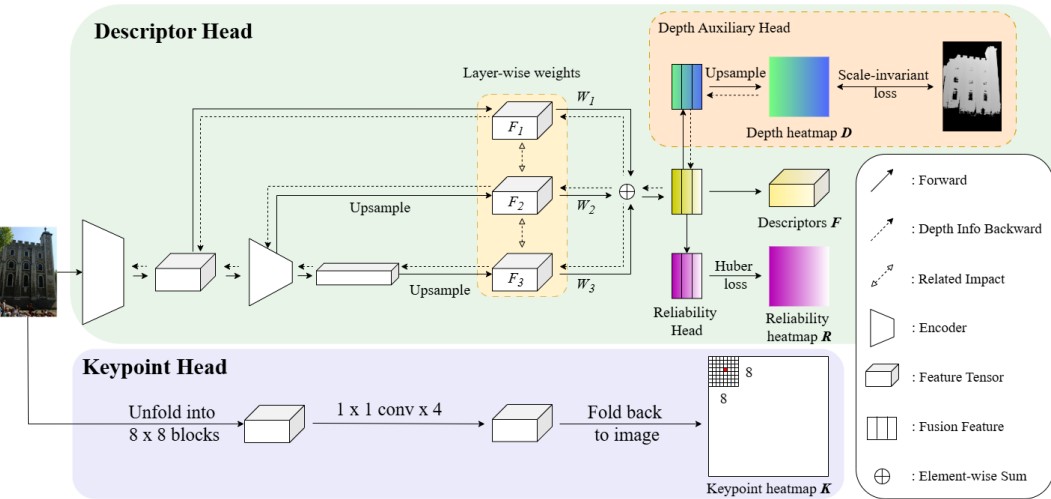

Figure 3: The DXFeat* architecture builds upon XFeat, with enhancements to feature extraction through learnable weights applied at different layers. A dedicated depth estimation branch is introduced in the Descriptor Head's fusion block. This branch uses a two-layer bottleneck followed by learnable upsampling to generate a depth map, which is then supervised by ground truth. This allows backpropagation to embed depth information into the feature representation, improving the quality of the extracted features.

and description. Recent advancements in transformer-based architectures (Sarlin et al., 2020; Lindenberger et al., 2023) have improved robustness and accuracy, but these methods often lead to high computational costs. In contrast, we show that it is possible to reduce compute usage significantly while maintaining performance. Efficient methods like SiLK and SuperPoint achieve comparable accuracy, but it has an advantage by simplifying keypoint extraction, despite the trade-off of slower inference. Other lightweight models, such as ZippyPoint (Kanakis et al., 2023), use optimizations like quantization, but their applicability is limited by hardware constraints. LightGlue (Lindenberger et al., 2023) offers faster matching but remains costly due to its transformer-based design. ALIKED (Zhao et al., 2023) simplifies keypoint detection and descriptor extraction for efficiency, while XFeat (Potje et al., 2024) offers a low-complexity solution with competitive performance. Our approach not only prioritizes efficiency but also significantly enhances robustness, making it particularly well-suited for deployment on simple devices without sacrificing performance.

**Lightweight Depth Estimation for Resource-Constrained Systems:** Recent work in lightweight depth estimation emphasizes reducing computation while preserving accuracy. FastDepth (Wofk et al., 2019) and Monodepth2 (Lin et al., 2015) introduce efficient architectures for real-time inference on constrained devices, while LiteDepth (Cui et al., 2021) further lowers cost through a two-stage framework with reduced input resolution. These approaches parallel our goal of combining efficient keypoint matching and depth estimation to enable real-time deployment on general-purpose hardware.

## 3 METHODOLOGY

The key distinction between DXFeat and XFeat Potje et al. (2024) is the introduction of a Relative Depth Consistency (RDC) Loss, which leverages depth-assisted information to enhance descriptor robustness under viewpoint-induced depth variations. Instead of enforcing strict depth invariance, RDC encourages the network to maintain consistent relative depth relationships between pixels, ensuring stable local descriptors and reliable keypoint recognition even when geometric distortions or focal length changes occur. This adaptation directly addresses the challenges that often degrade traditional descriptors in real-world scenarios.

To maintain a lightweight model, we introduce an auxiliary depth branch that supports network learning during training but is removed during inference to preserve efficiency. By incorporating

a specialized detection head and a depth-aware reliability loss, DXFeat integrates depth cues into backpropagation, improving resilience to depth-induced artifacts such as blur and perspective distortions.

We leverage depth maps from the MegaDepth dataset, enabling supervision across diverse scenes with significant viewpoint and depth variation. This provides a stable and accurate keypoint detection mechanism, making the framework particularly suitable for applications such as structure-from-motion and augmented reality, where feature reliability under depth changes is critical.

### 3.1 ARCHITECTURE

DXFeat builds upon the backbone architecture of XFeat (Potje et al., 2024), a lightweight framework that has demonstrated strong performance in semi-dense feature matching. Let $I \in \mathbb{R}^{H \times W \times 1}$ be a grayscale image, where $H$ is the height, $W$ the width in pixels. The model processes grayscale images as input and employs a distillation-based feature pyramid fusion strategy to construct compact and expressive descriptors. This approach effectively integrates multi-scale visual information, producing a more robust and discriminative representation. Unlike SuperPoint's (DeTone et al., 2018) weight-sharing mechanism, XFeat assigns keypoint detection to a dedicated branch, enabling more specialized processing and improving feature extraction efficiency.

While feature distillation effectively captures low-level visual cues, it often results in the loss of high-level semantic details (Gou et al., 2021; Phuong & Lampert, 2019; Habib et al., 2024). To address this, we introduce a layer-wise weight strategy, where each pyramid level has its own learnable weights. This enables the network to focus on the unique contributions of each feature level, improving descriptor stability and resilience across varying image conditions.

Mathematically, let $F_1, F_2, F_3$ represent the feature maps of the three layers, where $F_1 \in \mathbb{R}^{H/8 \times W/8 \times 64}$, $F_2 \in \mathbb{R}^{H/16 \times W/16 \times 64}$, and $F_3 \in \mathbb{R}^{H/32 \times W/32 \times 128}$. Each layer has an associated weight $W_1, W_2, W_3$, which can be learned independently. To facilitate multi-scale feature fusion, $F_2$ and $F_3$ undergo upsampling to match the spatial resolution of $F_1$, resulting in $\hat{F}_2, \hat{F}_3 \in \mathbb{R}^{H/8 \times W/8 \times 64}$. The final feature representation $\hat{F}$ is computed as:

$$\hat{F} = W_1 \cdot F_1 + W_2 \cdot \hat{F}_2 + W_3 \cdot \hat{F}_3, \tag{1}$$

where $\hat{F}_2 = \text{Upsample}(F_2)$ and $\hat{F}_3 = \text{Upsample}(F_3)$ are the upsampled output of $F_2$ and $F_3$, respectively. This formulation allows the network to effectively combine low-level and high-level features, with each layer contributing according to its learned weight, thereby generating a more robust and stable feature descriptor.

To enhance model performance, we introduce a lightweight depth auxiliary branch to XFeat's fusion, depth-aware training to improve robustness under viewpoint-induced depth changes. This branch comprises a 2D convolution, batch normalization, and activation, with a learnable upsampling convolution refining depth estimates while preserving spatial consistency. Unlike explicit depth estimation, our approach offers a coarse yet effective auxiliary estimation, maintaining descriptor consistency under varying depth conditions. Further details are provided in subsequent sections.

### 3.2 DEPTH AUXILIARY BRANCH

The depth auxiliary branch in DXFeat* is a novel integration of depth estimation and auxiliary learning, designed to enhance the robustness of local descriptors. Unlike prior methods such as Alike (Zhao et al., 2022), which introduced disparity loss but relied heavily on non-maximum suppression (NMS) modules, our approach ensures that depth information directly influences the feature learning process without excessive dependence on post-processing techniques. While Alike's disparity loss provides some benefits, it lacks a strong direct impact on descriptor learning, limiting its ability to generalize across varying depth conditions.

By leveraging a pyramid structure, the fused feature representation is obtained with dimensions $F \in \mathbb{R}^{H/8 \times W/8 \times 64}$. This representation is then processed through a series of convolutional layers, gradually upsampling and refining the depth information until it recovers the final depth prediction

$\boldsymbol{D} \in \mathbb{R}^{H \times W \times 1}$. This depth-aware module ensures robust feature learning across varying depth scales, thereby enhancing the model's performance in complex real-world scenarios.

We train DXFeat* in a supervised manner using pixel-level ground truth correspondences. Given an image-depth pair $(I_1, D_1)$, where $I_1$ represents the original image and $D_1$ is its corresponding depth map, we define a matching set between two images $(I_1, I_2)$ as $M_{I_1 \leftrightarrow I_2} \in \mathbb{R}^{N \times 4}$. Here, each row of $M_{I_1 \leftrightarrow I_2}$ represents a corresponding keypoint pair, where the first two columns contain the $(x, y)$ coordinates in $I_1$, and the last two columns contain the corresponding coordinates in $I_2$. The accuracy of these correspondences is influenced by the depth

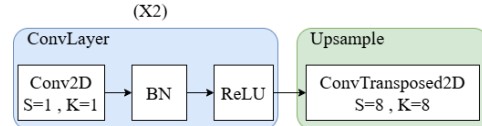

Figure 4: The depth auxiliary branch integrates depth cues into the original features via two convolutional layers and an upsampling layer.

similarity between $D_1$ and $D_2$, where a smaller disparity between them indicates better alignment and more reliable feature matching.

To enhance model performance, we introduce a depth auxiliary branch, extending XFeat's fusion process for relative-depth consistency. This lightweight branch, consisting of a 2D convolutional layer Conv, batch normalization BN, and an activation function $\sigma$, generates the depth map prediction $D$ from the feature map $F$. The depth map is generated through two iterations of $\mathcal{T}(\mathcal{T}(F))$, followed by a learnable upsampling using a $\mathrm{Conv}_2^T$ (transpose convolution) layer. Here, $\mathcal{T}$ represents the operation of applying batch normalization and activation function on the feature map, defined as

$$\mathcal{T}(F) = \sigma\left(\mathrm{BN}(\mathrm{Conv}(F))\right). \tag{2}$$

The final depth map $D$ is computed as

$$D = \mathrm{Conv}_2^T\left(\mathcal{T}\left(\mathcal{T}(F)\right)\right). \tag{3}$$

### 3.3 RELATIVE DEPTH CONSISTENCY LOSS

By embedding depth-aware cues into the feature representation, DXFeat* improves descriptor robustness under illumination changes, viewpoint-induced depth variations, and geometric distortions. This ensures that local descriptors remain consistent across images captured from different viewpoints or focal lengths, addressing a limitation overlooked in prior works.

To achieve this, we adopt a Relative Depth Consistency (RDC) Loss, inspired by the scale-invariant formulation of Eigen et al. (Eigen et al., 2014), but adapted to our setting. Unlike Eigen et al., who introduced a logarithmic term to compress large absolute depth ranges, our framework normalizes depth values within each scene to $[0, 1]$, representing relative depth rather than absolute metric depth. In this regime, applying a logarithm is unnecessary and could distort the normalized depth distribution. Removing the log allows the loss to directly measure relative depth differences, resulting in more stable optimization and better generalization, while still preserving the essential relative depth relationships.

The loss formulation is expressed as:

$$L_{\mathrm{RDC}} = \frac{1}{N} \sum_i (d_i - \hat{d}_i)^2 \ - \ \lambda_{\mathrm{RDC}} \left( \sum_i d_i - \sum_i \hat{d}_i \right)^2 \tag{4}$$

- The first term $\frac{1}{N} \sum_i (d_i - \hat{d}_i)^2$ penalizes pixel-wise discrepancies, emphasizing local relative depth cues.

- The second term $\lambda_{\mathrm{RDC}} \left( \sum_i d_i - \sum_i \hat{d}_i \right)^2$ enforces global consistency, mitigating bias from overall depth shifts while preserving the relative structure.

Here, $\lambda_{\mathrm{RDC}}$ balances the global consistency term. In our experiments, we set $\lambda_{\mathrm{RDC}} = 0.5$.

### 3.4 Reliability Map Optimization

The reliability loss in XFeat measures the consistency between predicted reliability maps and their ground truth. For each image pair, the ground truth reliability is computed from the similarity matrix: we first apply row-wise and column-wise softmax to the keypoints, then take the maxima along each dimension and multiply them to obtain a per-keypoint confidence score. The model predicts a reliability map, which is compressed to $[0, 1]$ via a sigmoid function, and the predictions $\sigma(R_1)$ and $\sigma(R_2)$ are trained to align with this ground truth.

Originally, XFeat used L1 loss to supervise the reliability map:

$$L_{\text{L1}}(R, R^*) = \sum_i |R_i - R_i^*|, \tag{5}$$

which penalizes absolute differences uniformly. While simple, L1 loss lacks fine-grained control for small errors and may lead to unstable gradient updates when the supervision signal exhibits high variance.

To address this, we adopt the Huber loss (Gokcesu & Gokcesu, 2021):

$$L_{\text{Huber}}(R, R^*) = \begin{cases} \frac{1}{2}(R - R^*)^2, & \text{if } |R - R^*| \leq \delta, \\ \delta\left(|R - R^*| - \frac{1}{2}\delta\right), & \text{otherwise}, \end{cases} \tag{6}$$

where $\delta$ controls the transition between L2 behavior for small residuals and L1 behavior for large residuals. In our experiments, $\delta = 0.1$. The primary motivation for using Huber loss is gradient stabilization: multi-modal supervision from relative depth introduces larger variance in predicted reliabilities, and L1 loss may produce unstable gradients for inlier pixels. Huber loss allows fine-grained updates when predictions are close to the ground truth, while preventing gradient explosion for larger deviations.

Thus, the Huber-based reliability loss is formulated as:

$$L_{\text{reliability}} = \sum_i L_{\text{Huber}}(R_i, R_i^*), \tag{7}$$

summing over all pixels. By combining L2 behavior for small errors with L1 behavior for large errors, the network learns more precise confidence estimates while maintaining stable training, resulting in improved feature matching performance across noisy or ambiguous scenarios.

## 4 Experiments

To verify the proposed approach, we implemented the model in PyTorch (Paszke et al., 2019) and adopted the same training and experimental configurations as XFeat. The model was trained on a dataset composed of MegaDepth (Li & Snavely, 2018) and synthetically warped COCO (Lin et al., 2015) images in a 6:4 ratio, and all images were resized to (W = 800, H = 600). Following this mixed-training strategy, enhances the model's generalization ability, aligning with prior research findings. The training process utilized the Adam optimizer (Kingma & Ba, 2014) with a batch size of 10 pair of images and involved a total of 160,000 parameter updates. Due to the inclusion of the depth auxiliary branch, training required slightly more time compared to the baseline model, extending to approximately 40 hours on an NVIDIA RTX 6000 Ada GPU.

For evaluation, the proposed method is benchmarked using the same protocol as XFeat and conducted comparisons with multiple existing approaches, including DISK (Tyszkiewicz et al., 2020), SiLK (Gleize et al., 2023), SuperPoint (DeTone et al., 2018), ZippyPoint (Kanakis et al., 2023), ALIKE (Zhao et al., 2022), LiftFeat (Liu et al., 2025), and ORB (Rublee et al., 2011). In cases where 10,000 keypoints were extracted for evaluation, these methods were marked with "*".

Table 1: Relative camera pose estimation on MegaDepth-1500. Our method surpasses other lightweight approaches, improving upon XFeat* by 3% under the strictest criterion and nearing DISK's performance. Speed tests show comparable runtime to XFeat after branch removal, with superior pose estimation. An asterisk (*) indicates 10k keypoints, and the best/second-best results (Standard/Fast) are in bold/underlined.

| Method | AUC@5° | AUC@10° | AUC@20° | Acc@10° | FPS(CPU) | FPS(GPU) |
|---|---|---|---|---|---|---|
| SiLK (Gleize et al., 2023) | 14.7 | 21.5 | 29.3 | 31.9 | 0.30 | 8.05 |
| SiLK* (Gleize et al., 2023) | 16.2 | 23.2 | 31.8 | 34.7 | 0.29 | 8.01 |
| SuperPoint (DeTone et al., 2018) | 37.3 | 50.1 | 61.5 | 67.4 | 1.36 | 17.79 |
| DISK (Tyszkiewicz et al., 2020) | 53.8 | 65.9 | 75.0 | 81.3 | 1.13 | 18.03 |
| DISK* (Tyszkiewicz et al., 2020) | **55.2** | 66.8 | 75.3 | 81.3 | 1.13 | 18.03 |
| ORB (Rublee et al., 2011) | 17.9 | 27.6 | 39.0 | 43.1 | **66.89** | X |
| ZippyPoint (Kanakis et al., 2023) | 23.6 | 34.9 | 46.3 | 51.8 | 1.32 | 19.35 |
| ALIKE (Zhao et al., 2022) | 49.4 | 61.8 | 71.4 | 77.7 | 2.85 | 30.34 |
| XFeat (Potje et al., 2024) | 42.6 | 56.4 | 67.7 | 74.9 | 10.80 | 125.83 |
| XFeat* (Potje et al., 2024) | 50.4 | 65.8 | 77.5 | 85.1 | 7.45 | 48.17 |
| LiftFeat (Liu et al., 2025) | 44.7 | 59.5 | 70.3 | 77.5 | 6.10 | 95.3 |
| DXFeat(ours) | 42.5 | 56.9 | 68.4 | 75.9 | 10.84 | **125.91** |
| DXFeat*(ours) | 53.2 | **68.6** | **80.1** | **88.3** | 7.39 | 47.30 |

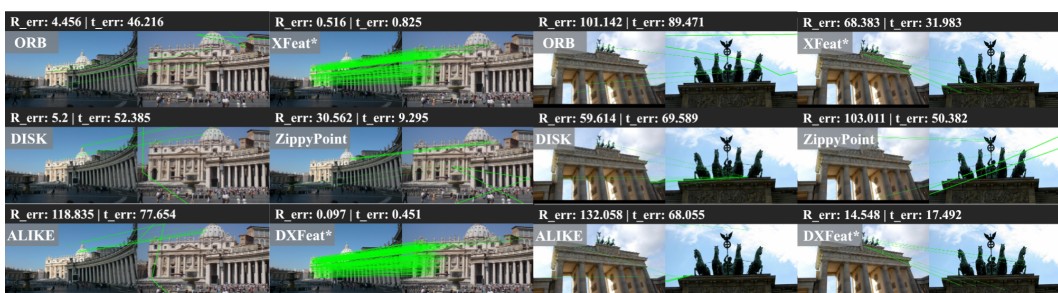

Figure 5: Results on MegaDepth-1500. We specifically identified both simple and challenging scenes to evaluate rotation and translation error estimation. The results clearly indicate that, in challenging scenarios, other methods fail to demonstrate any significant advantage, whereas DXFeat consistently exhibits the smallest error estimates, with the difference being highly significant—making it the best-performing method. In simpler scenes, while other methods also show effectiveness, our framework still maintains the smallest error among all approaches, further highlighting its robustness and superiority across a wide range of conditions.

## 4.1 RELATIVE POSE ESTIMATION

**Dataset and Preprocessing:** The proposed approach is evaluated on the MegaDepth (Li & Snavely, 2018) and ScanNet (Dai et al., 2017) test sets, following the same protocol as previous studies. These datasets include scenes with significant viewpoint and illumination variations, as well as repetitive structures, making them particularly challenging. The camera poses provided in these datasets do not overlap with our training data. For essential matrix estimation, we employ LO-RANSAC (Larsson & contributors, 2020). To ensure a fair comparison across different methods, we optimize the threshold settings individually. Images from MegaDepth are resized to a maximum dimension of 1,200 pixels, while ScanNet images are kept at their default VGA resolution(480×640).

**Evaluation Metrics:** In this experiment, different thresholds of Area Under the Curve (AUC) and Accuracy (ACC) at 5°, 10°, and 20° are used to evaluate the performance of relative pose estimation within various angular error ranges. Additionally, the frames per second (FPS) are measured on both an Intel Core i7-12700K CPU and an NVIDIA RTX 3060 Ti GPU, providing a comprehensive assessment of the system's predictive accuracy and computational efficiency.

Table 2: ScanNet-1500 relative pose estimation. Compared to XFeat*, DXFeat* demonstrates superior generalization performance in indoor scenes.

| Method | AUC@5° | AUC@10° | AUC@20° |
|---|---|---|---|
| SuperPoint | 12.5 | 24.4 | 36.7 |
| DISK | 9.6 | 19.3 | 30.4 |
| DISK* | 11.3 | 22.3 | 33.9 |
| ORB | 9.0 | 18.5 | 29.9 |
| ALIKE | 8.0 | 16.4 | 25.9 |
| XFeat | 16.7 | 32.6 | 47.8 |
| XFeat* | _18.5_ | 34.4 | 49.6 |
| LiftFeat | _18.5_ | _34.9_ | _51.2_ |
| DXFeat | 17.6 | 33.4 | 48.6 |
| DXFeat* | **19.6** | **36.5** | **52.3** |

Table 3: Homography estimation on HPatches dataset. DXFeat achieves high-quality results with similar computational overhead.

| Method | Illumination MHA | | | Viewpoint MHA | | |
|---|---|---|---|---|---|---|
| | @3 | @5 | @7 | @3 | @5 | @7 |
| SiLK | 78.5 | 82.3 | 83.8 | 48.6 | 59.6 | 62.5 |
| SuperPoint | 94.6 | 98.5 | 99.8 | 71.1 | 79.6 | 83.9 |
| DISK | 94.6 | 98.8 | 99.6 | 66.4 | 77.5 | 81.8 |
| ORB | 74.6 | 84.6 | 85.4 | 63.2 | 71.4 | 78.6 |
| ZippyPoint | 94.2 | 96.9 | 98.5 | 66.1 | 76.8 | 80.7 |
| ALIKE | 94.6 | _98.5_ | **99.6** | 68.2 | 77.5 | 81.4 |
| XFeat | 95.0 | 98.1 | 98.8 | 68.6 | 81.1 | **86.1** |
| XFeat* | 93.5 | 98.1 | 98.9 | 50.4 | 74.6 | 82.9 |
| LiftFeat | **95.6** | **98.8** | 99.2 | **71.1** | 81.7 | **87.5** |
| DXFeat | _95.3_ | 97.6 | 98.9 | _69.6_ | 79.3 | 85.0 |
| DXFeat* | 95.1 | 97.7 | 98.9 | 68.9 | **82.1** | **87.5** |

**Results and Comparison:** In the Camera Relative Pose task, we evaluate AUC and accuracy at different strictness levels (5°, 10°, 20°). As shown in the Table 1, DXFeat* achieves consistent improvements of 2.6% to 3.2% while maintaining similar speed. To assess generalization, we also use ScanNet-1500 Table 2, where DXFeat* achieves top performance at all strictness levels, surpassing XFeat* by 1.1% to 2.7% in AUC and demonstrating the robustness of our approach.

### 4.2 HOMOGRAPHY ESTIMATION

**Dataset and Preprocessing:** The proposed approach is evaluated on the widely used HPatches (Balntas et al., 2017) dataset, which consists of image sequences from planar scenes that exhibit varying degrees of viewpoint and illumination changes. To ensure robust homography estimation across different methods, we employ MAGSAC++ (Barath et al., 2020), a well-established technique for handling outliers when computing transformations from keypoint correspondences.

**Evaluation Metrics:** Following the protocol used in ALIKE, the performance is assessed using Mean Homography Accuracy (MHA). This metric is computed based on the average corner error in pixels, where the four corners of the reference image are projected onto the target image using both the ground-truth and estimated homographies. Accuracy is reported at predefined thresholds of 3, 5, and 7 pixels.

**Results and Comparison:** Due to the minimal viewpoint variation in the Illumination subset, the evaluation complexity remains relatively low, resulting in consistently high MHA scores across most models. This suggests that under stable lighting conditions, existing methods effectively extract and match features. In contrast, the Viewpoint subset presents a greater challenge, as it includes image pairs with substantial viewpoint differences, leading to perspective distortion and scale variations that hinder feature matching.As shown in Table 3 DXFeat* significantly outperforms other methods in Viewpoint MHA, demonstrating its superior ability to handle extreme viewpoint changes. This indicates that DXFeat* is more effective in learning better generalization under viewpoint changes, likely due to its architectural design or training strategy. The performance gap highlights the importance of developing feature extraction models with improved robustness to geometric transformations.

### 4.3 ABLATION

Our ablation study investigates the impact of three key modifications in DXFeat*: (i) layer-wise weighting (L), (ii) a robust Huber loss function (H), and (iii) a depth auxiliary branch (D).

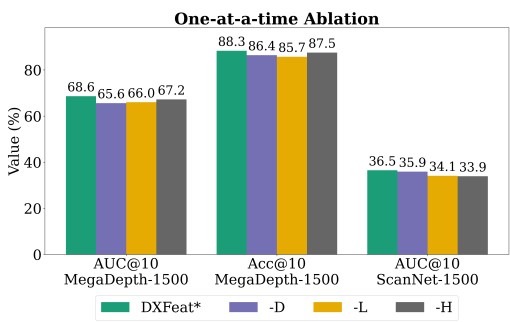

Figure 6: One-at-a-time ablation visualization.

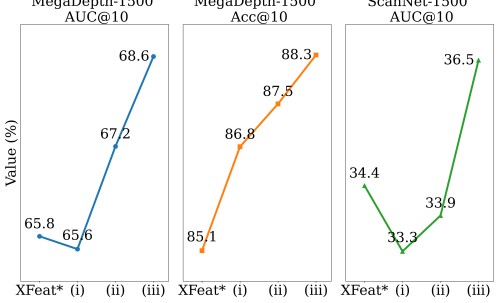

Figure 7: Visualization of incremental ablation.

Experimental results confirm that each of these components plays a crucial role in performance improvement.

**The one-at-a-time ablation:** Table 4 and Figure 6 highlight the significance of these modifications.Removing D leads to the most substantial performance drop, with AUC@10 decreasing by 3.0% on MegaDepth-1500 and 0.6% on ScanNet-1500, underscoring its role in enhancing feature representation. Excluding L results in a 2.6% decline in Acc@10, emphasizing its importance for feature matching accuracy.

**The incremental integration:** Table 5 and Figure 7 further validate these findings.

While H has a smaller impact, it still contributes to overall stability by improving loss function robustness. Adding D initially increases Acc@10 by 1.7%, though AUC@10 sees a slight decline (-0.2% on MegaDepth-1500, -1.1% on ScanNet-1500), indicating that its effectiveness is maximized when combined with other enhancements. Incorporating L leads to additional gains in both AUC@10 and Acc@10, reinforcing its contribution to feature matching. Finally, integrating all three components (D, L, H) into DXFeat* achieves the best results, with AUC@10 improving by 2.8%, Acc@10 by 3.2%, and ScanNet-1500 performance increasing by 2.1%, demonstrating their complementary effects.

In summary, these ablation studies (Tables 4 and 5) confirm the individual and combined contributions of D, L, and H, showing that their integration leads to a significant performance boost in DXFeat*.

Table 4: Ablation study on module reduction. Removing each module significantly reduces performance.

| Method | MegaDepth | | ScanNet |
| --- | --- | --- | --- |
| | AUC@10° | Acc@10° | AUC@10° |
| DXFeat* | 68.6 | 88.3 | 36.5 |
| (i) -D | 65.6 (↓3.0) | 86.4 (↓1.9) | 35.9 (↓0.6) |
| (ii) -L | 66.0 (↓2.6) | 85.7 (↓2.6) | 34.1 (↓2.4) |
| (iii) -H | 67.2 (↓1.4) | 87.5 (↓0.8) | 33.9 (↓2.6) |

Table 5: Incremental Ablation. Performance impact of progressively adding D, L, and H strategies to XFeat.

| Method | MegaDepth | | ScanNet |
| --- | --- | --- | --- |
| | AUC@10° | Acc@10° | AUC@10° |
| XFeat* | 65.8 | 85.1 | 34.4 |
| (i) +D | 65.6 (↓0.2) | 86.8 (↑1.7) | 33.3 (↓1.1) |
| (ii) +D, L | 67.2 (↑1.4) | 87.5 (↑2.4) | 33.9 (↓0.5) |
| (iii) +D, L, H | 68.6 (↑2.8) | 88.3 (↑3.2) | 36.5 (↑2.1) |

## 5 CONCLUSION

We present a lightweight image-matching framework with a depth auxiliary branch. Depth-guided training improves accuracy while reducing compute, and experiments/ablations show robustness across datasets and resolutions, enabling real-time use on mobile/embedded devices. The approach suits visual localization, AR, and robotics, balancing performance and cost.

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
