# A    BASELINE METHOD

## A.1    NETWORK TRAINING

We train DXFeat in a supervised manner using pixel-level ground-truth correspondences. Given an image pair $(I_1, I_2)$, we assume the availability of $N$ matched pixels, denoted as:

$$M_{I_1 \leftrightarrow I_2} \in \mathbb{R}^{N \times 4}. \tag{8}$$

The first two columns of $M_{I_1 \leftrightarrow I_2}$ represent the $(x, y)$ coordinates in $I_1$, while the last two columns provide the corresponding coordinates in $I_2$.

## A.2    LEARNING LOCAL DESCRIPTORS

To train the local feature embeddings $F$, we employ a negative log-likelihood (NLL) loss. Two sets of descriptors, $F_1$ from image $I_1$ and $F_2$ from image $I_2$, are extracted from the dense feature maps $F(\cdot, \cdot)$, where each is represented as $\mathbb{R}^{N \times 64}$. Each row $F_1(i, \cdot)$ and $F_2(i, \cdot)$ corresponds to a pair of 64-dimensional descriptors from the same pixel in $I_1$ and $I_2$, respectively.

The similarity matrix $S \in \mathbb{R}^{N \times N}$ is computed as $S = F_1 F_2^T$. To enforce bidirectional consistency in feature matching, we employ the dual-softmax loss $L_{ds}$, ensuring that corresponding features are emphasized along the main diagonal $S_{ii}$:

$$L_{ds} = -\sum_i \log\left(\text{softmax}_r(S)_{ii}\right) - \sum_i \log\left(\text{softmax}_r(S^T)_{ii}\right). \tag{9}$$

## A.3    LEARNING RELIABILITY MAPS

To estimate confidence in the feature matching process, we derive a reliability map using the dual-softmax approach:

$$\bar{R}_1 = \max_r(\text{softmax}_r(S)), \quad \bar{R}_2 = \max_r(\text{softmax}_r(S^T)). \tag{10}$$

These scores provide a measure of feature distinctiveness and the likelihood of successful matching. To enforce robust confidence estimation, we replace the original $L_1$ loss with the Huber loss:

$$L_{\text{Huber}}(R, R^*) = \begin{cases} \frac{1}{2}(R - R^*)^2, & \text{if } |R - R^*| \le \delta, \\ \delta ||R - R^*| - \frac{1}{2}\delta|, & \text{otherwise.} \end{cases} \tag{11}$$

## A.4    LEARNING PIXEL OFFSETS

To refine feature matching precision, we introduce pixel-level offset supervision. The Match Refinement Module is trained using ground-truth offsets $(\bar{x}, \bar{y})$ derived from $M_{I_1 \leftrightarrow I_2}$. We employ NLL loss over the logits $o$ to optimize fine matching:

$$L_{\text{fine}} = -\sum_i \log\left(\text{softmax}(o_i)\right)_{\bar{y}_i, \bar{x}_i}. \tag{12}$$

## A.5    LEARNING KEYPOINTS

The keypoint detection module is minimalist, relying on knowledge distillation from a larger teacher network instead of handcrafted keypoint loss functions. We use ALIKE keypoints obtained from a tiny backbone as supervision.

Given the keypoint probability logit map $K \in \mathbb{R}^{H/8 \times W/8 \times (64+1)}$, keypoint locations $(t_x, t_y)$ are mapped to linear indices as $t_{\text{idx}} = (t_x + t_y \times 8)$, where $t_{\text{idx}} \in \{0, 1, \dots, 63\}$. When no keypoint is detected within a given cell $k_{i,j}$, we set $t_{\text{idx}} = 64$.

To avoid class imbalance during training, we cap the number of negative (non-keypoint) samples. The final NLL loss for keypoint detection is:

$$L_{kp} = -\sum_k \log\left(\text{softmax}(k_{i,j})\right)_{t_{\text{idx}}}. \tag{13}$$

## A.6 LEARNING DEPTH INFORMATION

To integrate depth cues into feature extraction, we introduce a scale-invariant depth loss $L_{SI}$:

$$L_{\text{RDC}} = \frac{1}{N} \sum_i (d_i - \hat{d}_i)^2 \; - \; \lambda_{\text{RDC}} \left( \sum_i d_i - \sum_i \hat{d}_i \right)^2 \tag{14}$$

The first term represents the Mean Squared Error (MSE) between predicted depth and ground truth. The second term enforces scale invariance, ensuring the network learns relative depth relationships rather than absolute depth values.

## A.7 FINAL LOSS FUNCTION

Combining all components, the final loss function is defined as:

$$L = \alpha L_{ds} + \beta L_{\text{Huber}} + \gamma L_{\text{fine}} + \delta L_{kp} + \eta L_{RDC}. \tag{15}$$

where the weights are set to specific values: $\alpha = 1$, $\beta = 30$, $\gamma = 2$, $\delta = 1$, and $\eta = 1$. These values ensure a balanced contribution of each term to the overall loss function.

## B FIXED VS. LEARNABLE PYRAMID FUSION WEIGHTS

To effectively integrate multi-scale features, we compared two strategies for pyramid fusion. Our final design uses **globally learned, then frozen** layer-wise weights: $\bar{W}_1, \bar{W}_2, \bar{W}_3$ that are optimized jointly with the network during training and then **frozen** at inference time:

$$\hat{F} \; = \; W_1 F_1 \; + \; W_2 \hat{F}_2 \; + \; W_3 \hat{F}_3.$$

## B FIXED VS. LEARNABLE PYRAMID FUSION WEIGHTS

We compare two strategies for multi-scale pyramid fusion. Our final design uses **globally learned, then frozen** layer-wise weights: three scalars $\bar{W}_1, \bar{W}_2, \bar{W}_3$ are optimized end-to-end with the network and kept fixed for all inputs at inference:

$$\hat{\mathbf{F}} \; = \; W_1 \mathbf{F}_1 \; + \; W_2 \hat{\mathbf{F}}_2 \; + \; W_3 \hat{\mathbf{F}}_3,$$

where $\hat{\mathbf{F}}_2$ and $\hat{\mathbf{F}}_3$ are the upsampled versions of $\mathbf{F}_2$ and $\mathbf{F}_3$ to match the spatial resolution of $\mathbf{F}_1$. Under our training protocol these scalars converge to $\bar{W}_1{=}0.45$, $\bar{W}_2{=}0.37$, $\bar{W}_3{=}1.72$—*learned, not hand-tuned*. Convolutional filters determine how features are extracted at each level, while these scalars control how much each level contributes to the fused descriptor. This explicit, globally learned weighting consistently outperforms naïve averaging and implicit convolutional fusion, yielding stable gains and better generalization in our ablations.

Table 6: Qualitative results. $*$ = semi-dense; others are sparse.

| Method | MegaDepth | | ScanNet |
|---|---|---|---|
| | AUC@10° | Acc@10° | AUC@10° |
| (i) XFeat | 56.4 | 74.9 | 32.6 |
| (ii) XFeat* | 65.8 | 85.1 | 34.4 |
| (iii) DXFeat | 56.9 | 75.9 | 33.4 |
| (iv) DXFeat* | 68.6 | 88.3 | 36.5 |
| (v) XFeat* + Conv-L | 65.1 | 84.3 | 34.0 |

For comparison, we implement an **input-conditioned (dynamic)** variant (Fig. 8) that predicts per-image weights $W_\ell(x)$ via a small Conv+GAP+FC head, adapting the fusion to each input at both training and inference. Despite this additional flexibility, accuracy drops noticeably (Tab. 6(v)). We hypothesize that dynamic per-image weights overfit short-term feature statistics and introduce optimization instability, degrading matching quality and confidence reliability.

Overall, the fixed-weight scheme offers a better robustness–accuracy trade-off in our setting; therefore we adopt the **globally learned–then–frozen** scalars as the default design in all subsequent experiments.

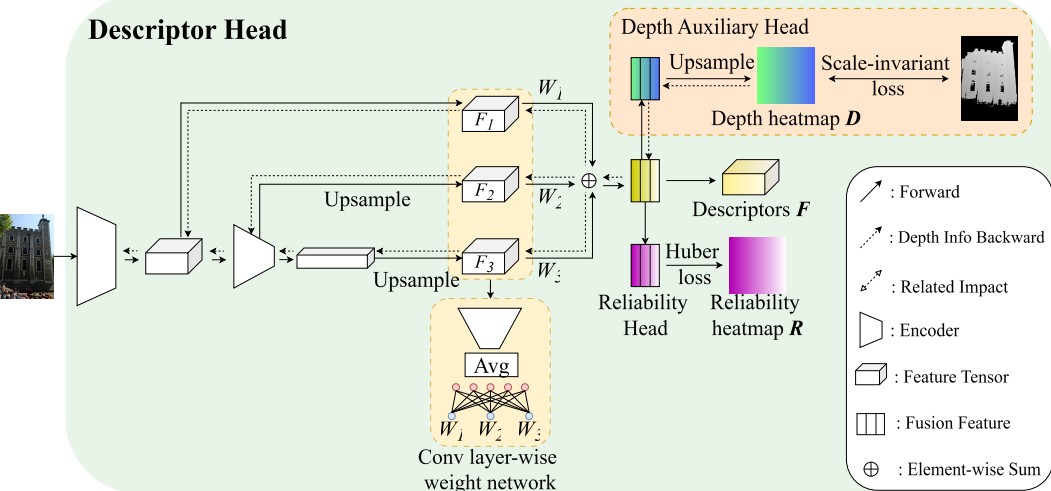

Figure 8: Illustration of the learnable weight fusion module.The network predicts weights for each pyramid level via Conv + GAP + FC.

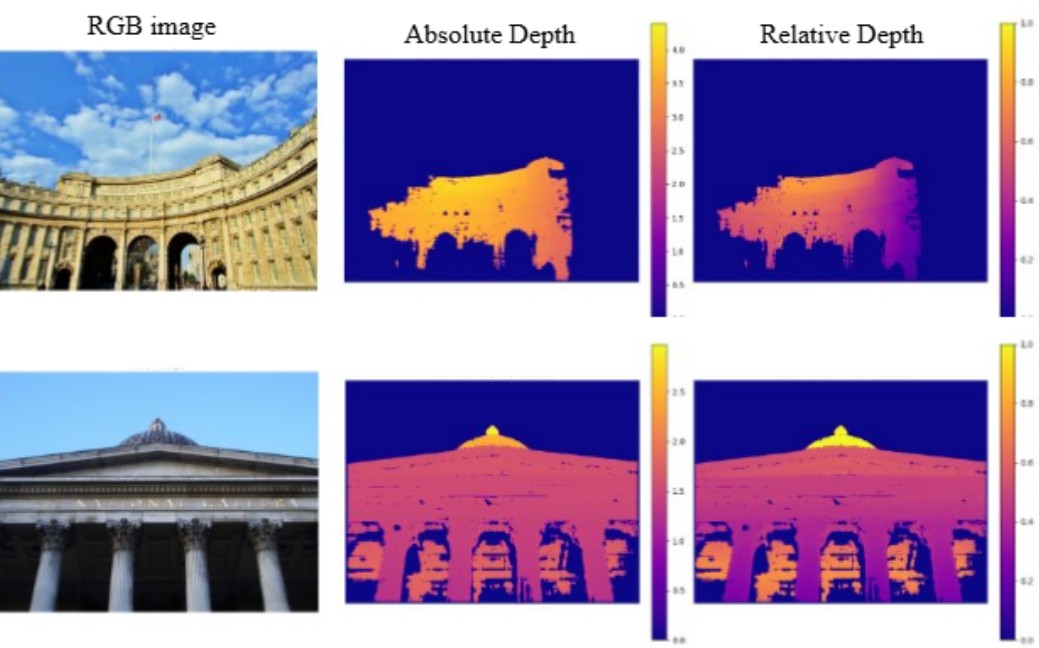

Figure 9: Depth normalization process. This procedure suppresses outliers, preserves structural gradients, and improves robustness for cross-view feature matching.

## C  DEPTH NORMALIZATION PROCESS

To enable our depth-aware loss function to learn *relative* depth distributions rather than relying solely on absolute depth values, we applied an additional normalization procedure to the MegaDepth dataset (Li & Snavely, 2018). Specifically, our goal was to map depth values from diverse scenes into a unified range of $[0, 1]$, allowing the model to focus on capturing structural depth gradients while mitigating biases caused by absolute scale differences.

For each scene, we first computed the mean $\mu$ and standard deviation $\sigma$ of depth values. Each pixel depth $d$ was then processed through a clamping operation:

$$d_{\text{clamp}} = \min\big(\max(d, \mu - 3\sigma),\ \mu + 3\sigma\big) \tag{16}$$

which ensures that all pixel values are restricted within the confidence interval $[\mu - 3\sigma,\ \mu + 3\sigma]$. This step effectively suppresses extreme outliers, often introduced by noise or sensor artifacts. Subsequently, the clamped depths were normalized into the range $[0, 1]$:

$$d_{\text{norm}} = \frac{d_{\text{clamp}} - (\mu - 3\sigma)}{(\mu + 3\sigma) - (\mu - 3\sigma) + \epsilon} \tag{17}$$

where $\epsilon$ is a small constant to prevent division by zero. In practice, every single pixel depth value in the map undergoes this two-step transformation (clamping followed by normalization), ensuring stable and consistent training signals across heterogeneous scenes.

This process enhances the stability and reliability of descriptor learning, enabling the model to emphasize depth gradients and structural consistency across viewpoints. Furthermore, aligning relative depth scales across multiple views of the same scene reduces biases from near-field or far-field depth variations, thereby improving robustness in cross-view and scale-sensitive feature matching. An example of this normalization pipeline is illustrated in Figure 9, where absolute depth maps are transformed into relative representations that better preserve scene geometry.

## D ABLATION ON SCALE-INVARIANT LOSS: WITH VS. WITHOUT LOG

A key distinction between our Relative Depth Consistency (RDC) loss and the scale-invariant loss of Eigen *et al.* Eigen et al. (2014) is whether a logarithmic transformation is applied to the depth domain. Eigen operated in $\log$ space to compress large depth ranges, but in our framework each scene's depth is already normalized into $[0, 1]$, making the $\log$ unnecessary and potentially harmful. Within this bounded range, $\log$ can distort the relative distribution and amplify gradients near zero, whereas the no-log formulation directly preserves relative depth relationships, yielding more stable optimization and better generalization.

Table 7: Ablation on RDC loss: with vs. without $\log$ transformation.

| Variant | MegaDepth-1500 | | | |
|---|---|---|---|---|
| | AUC@5° | AUC@10° | AUC@20° | Acc@10° |
| RDC (log) | 42.8 | 58.9 | 71.9 | 80.2 |
| RDC (no-log) | 51.5 | 65.0 | 78.6 | 86.7 |

The results suggest that applying $\log$ after per-scene normalization introduces training instability and reduces accuracy, while our no-log RDC consistently achieves higher performance. This validates our design choice of discarding the logarithmic term in the depth-aware consistency loss.

## E SCANNET VISUALIZATION

To further validate the robustness of DXFeat in indoor environments, we present qualitative results on the ScanNet dataset (Dai et al., 2017). As shown in Figure 10, DXFeat* consistently extracts more reliable keypoints and achieves more accurate correspondences compared to XFeat*.

In relatively simple cases with moderate viewpoint and illumination changes (Figure 10, top rows), DXFeat* achieves significantly lower rotation and translation errors than XFeat*, demonstrating improved stability under standard conditions. In more challenging scenarios, such as dynamic blur or larger viewpoint differences (Figure 10, bottom rows), DXFeat* leverages relative depth consistency to maintain robust matches, whereas XFeat* often produces erroneous or unstable correspondences.

These results highlight the superior performance of DXFeat* in handling both easy and difficult scenarios within indoor scenes, confirming its strong generalization ability and practical applicability in real-world visual localization tasks.

## F APPLICATION: VISUAL ODOMETRY (VO)

Visual Odometry (VO) estimates the camera trajectory from sequential image frames and has become a critical downstream task highly dependent on image matching quality. Feature correspon-

Figure 10: Visualization on ScanNet. Top: cases with moderate viewpoint and illumination changes, where DXFeat* achieves lower errors than XFeat*. Bottom: challenging cases with dynamic blur or large viewpoint differences, where DXFeat* still provides more reliable matches. Rotation ($R_{\text{err}}$) and translation errors ($t_{\text{err}}$) are reported for quantitative comparison.

dences extracted from consecutive frames provide the geometric constraints necessary for motion estimation, directly influencing the accuracy and stability of pose recovery.

Traditional approaches, such as ORB and SIFT, rely on handcrafted keypoints and descriptors, while recent learning-based methods aim to improve robustness under challenging conditions such as low texture, illumination changes, or motion blur. The typical VO pipeline consists of four main steps:
(1) feature detection and description.
(2) feature matching between adjacent frames using nearest-neighbor search and ratio test.
(3) relative pose estimation via Essential Matrix and RANSAC-based inlier selection.
(4) global pose recovery using Perspective-n-Point (PnP).
Reliable feature matching and robust motion estimation form a closed feedback loop, where VO both depends on and validates the quality of feature correspondences.

To evaluate the proposed DXFeat method in a real-world setting, we collected continuous image sequences along a walking trajectory in an outdoor environment with varying illumination and viewpoint changes. Figure 11 shows the ground-truth walking path (left) and representative image samples captured along the route (right).

Figure 12 presents trajectory reconstructions using ORB, SIFT, and DXFeat features. ORB exhibits noticeable drift and instability, while SIFT provides a smoother trajectory but still suffers from deviations at challenging segments. In contrast, DXFeat demonstrates the most consistent trajectory, closely aligning with the ground-truth path. These results confirm that DXFeat not only improves feature matching performance but also enables more stable and accurate VO in complex outdoor environments, highlighting its potential for downstream applications in robotics and autonomous navigation.

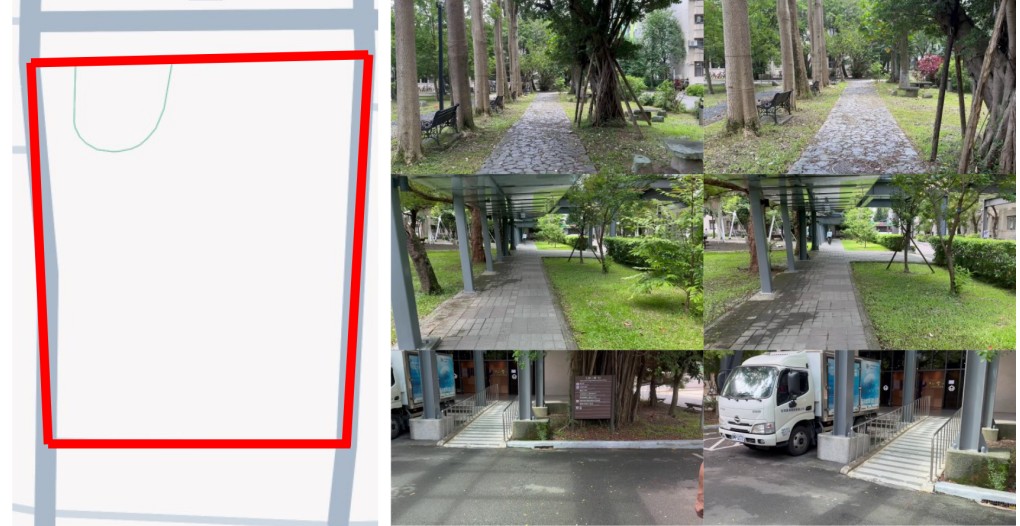

Figure 11: Ground-truth walking path (left, red polygon) and consecutive image frames (right) captured along the route for VO evaluation.

ORB                    SIFT                    DXFeat

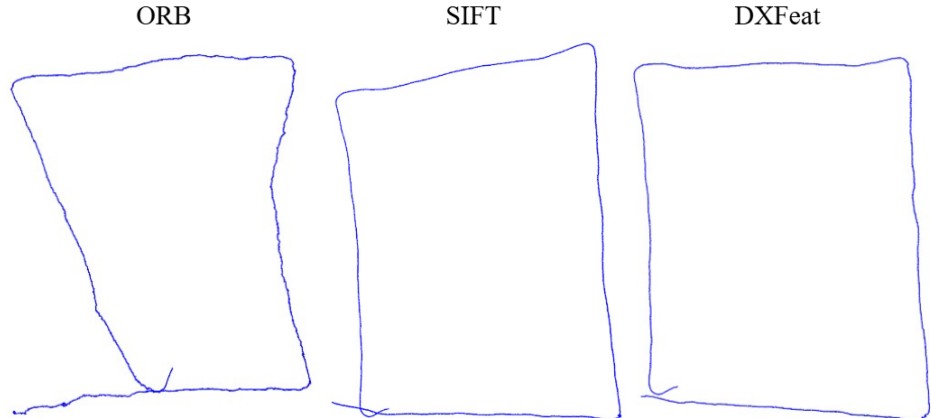

Figure 12: Visual Odometry trajectory comparison using ORB (left), SIFT (middle), and DXFeat (right). DXFeat produces a trajectory most consistent with the ground-truth path, demonstrating superior robustness and stability for VO.