# OpenReview forum: "DXFeat: Depth-Aware Features for Robust Image Matching"
_ICLR.cc/2026/Conference — Submitted to ICLR 2026_

### Official Review · Reviewer_puJS · 2025-10-22

**Soundness:** 2
**Presentation:** 2
**Contribution:** 2
**Rating:** 4
**Confidence:** 2

**Summary:**

This paper introduces DXFeat, a depth-aware feature extraction architecture designed for robust and efficient image matching. The core innovation is the integration of a depth auxiliary branch (used only in training) to enhance keypoint detection and local descriptors, paired with a modified reliability loss and layerwise learnable weight fusion. The method builds on the XFeat backbone but introduces a Relative Depth Consistency (RDC) loss and lightweight, learnable fusion of depth cues. DXFeat is evaluated across MegaDepth, ScanNet, and HPatches datasets, showing improved matching accuracy and runtime competitive with state-of-the-art lightweight methods. Both ablation studies and results tables suggest consistent performance boosts, supporting the case for its practical merits in resource-constrained settings.

**Strengths:**

The motivation for leveraging depth as an auxiliary training signal is clearly articulated, addressing practical problems such as keypoint loss from viewpoint changes and camera motion. DXFeat’s focus on real-world constraints (computational cost, mobile deployment, and varying scene conditions) is justified.

**Weaknesses:**

1. While the paper positions itself as improving XFeat by adding a depth auxiliary branch, it gives very limited discussion of other works that fuse depth or multimodal cues for detection/description. Additionally, the depth auxiliary branch proposed in this paper has not been validated in studies beyond XFeat. This weakens the argument for novelty, as very similar paradigms exist, albeit not always for exactly the same setting. Explicit experimental and conceptual differentiation is required.
2. The method in Table 2.3 should add the reference.
3. The manuscript does not acknowledge nor measure potential downsides, such as the costs of auxiliary depth supervision (dataset requirements, annotation effort) or failure cases when deploying on data where depth is unavailable at training.

**Questions:**

See weakness.

---

### Official Review · Reviewer_6Mbb · 2025-10-22

**Soundness:** 3
**Presentation:** 3
**Contribution:** 2
**Rating:** 4
**Confidence:** 4

**Summary:**

This paper introduces DXFeat, a depth-aware local feature extraction framework designed to improve keypoint detection and descriptor robustness in image matching tasks. Building on the XFeat architecture, the authors incorporate three main innovations: a depth auxiliary branch used only during training, a Relative Depth Consistency (RDC) loss that enforces geometric stability, and a Huber-based reliability loss to achieve smoother optimization. These additions are intended to make the model more resilient to depth and viewpoint variations while keeping inference efficiency on par with lightweight baselines. The method is evaluated on MegaDepth, ScanNet, and HPatches, where it achieves around 3% improvement over XFeat and performs competitively with other efficient local feature methods such as SiLK and ALIKE. Overall, the paper targets the important trade-off between robustness and efficiency in feature matching and provides solid empirical evidence of improvement, though the gains are modest.

**Strengths:**

1. The auxiliary depth branch used only during training is an elegant way to leverage geometric cues without inflating inference-time complexity.

2. Evaluation on three benchmarks (MegaDepth, ScanNet, HPatches) demonstrates robustness across indoor and outdoor settings.

3. DXFeat achieves consistent accuracy improvements while retaining inference speed comparable to XFeat.

4. The paper is generally well written, with clear motivation and consistent methodology organization.

**Weaknesses:**

1. The paper assumes availability of depth maps during training, but this assumption is restrictive for many real-world settings. The authors should discuss how DXFeat could be applied with synthetic or estimated depth (e.g., using foundation models like DepthAnything v2). An empirical comparison using such pseudo-depth could validate broader applicability.

2. The ablation on the auxiliary depth branch is shallow. There is no investigation into how branch structure, complexity, or depth quality affects final performance. Moreover, hyperparameter choices (e.g., λ_RDC = 0.5) are not justified or explored through sensitivity analysis.

3. The paper introduces multiple new loss terms (RDC and Huber-based reliability loss), but there is no visualization or trajectory analysis to explain how these affect convergence or stability. Such insights could strengthen understanding of why the model performs better.

4. The claim that depth supervision improves robustness is not empirically validated. It would be important to test DXFeat under perturbations or corruptions (e.g., Gaussian noise, blur, viewpoint jitter) to confirm its resilience.

5. The work is largely empirical. The authors provide no theoretical reasoning or qualitative evidence explaining why integrating depth improves descriptor generalization or how different modules interact. Visualizations (e.g., weight distributions, reliability maps, or feature embeddings) would add interpretability.

6.  Table 1 lacks critical computational comparisons, i.e., parameter counts, FLOPs, and inference latency. These metrics are essential for supporting claims about efficiency and lightweight deployment.

7. The ablation study (Tables 4–5) reports performance deltas but provides no deeper explanation of how the proposed components complement each other. More rigorous or incremental ablation analysis would clarify interdependencies between RDC, reliability loss, and layer-wise weighting.

8. Results are summarized without meaningful interpretation. For instance, the method fails to outperform baselines in HPatches illumination MHA but outperforms in viewpoint MHA, yet no reasoning is offered. Analytical discussion of such discrepancies would make the contribution more convincing.

9. Presentation issues. (1) Minor typos (e.g., missing spaces) and notation inconsistencies (e.g., variable “i” in Eq. 5) need correction.
(2) Figures (e.g., Fig. 4–7) lack sufficient explanation, particularly regarding variance and normalization.

**Questions:**

1. How would DXFeat perform if trained using synthetic or predicted depth maps from a foundation model (e.g., DepthAnything v2)?

2. How sensitive is the model to the λ_RDC hyperparameter and the structure of the depth branch?

3. Why was Huber loss preferred over other robust losses? Could the authors visualize its impact on training stability?

4. Can the authors report computational cost (FLOPs, parameters, latency) for fair comparison with other lightweight matchers?

5. How does DXFeat perform under input noise, viewpoint perturbation, or blur?

6. Why does DXFeat underperform in illumination-variant HPatches scenes, and how might this limitation be mitigated?

---

### Official Review · Reviewer_QZLU · 2025-10-26

**Soundness:** 3
**Presentation:** 3
**Contribution:** 2
**Rating:** 2
**Confidence:** 5

**Summary:**

The paper adds a depth-assisted regularization loss to XFeat, which helps in relative pose and homography estimation. The idea is quite straightforward, but more experiments would be needed to really prove its effectiveness.

**Strengths:**

The idea is straightforward and easy to follow.

**Weaknesses:**

A simple but truly effective idea can be ICLR-worthy. But right now the experiments don’t hit that bar.

Missing experiments:

(1) DeDoDe v2 is mentioned in related work, but there’s no comparison. Please add it.

(2) Check what DeDoDe v2 and XFeat typically evaluate on, and include those (e.g., Image Matching Challenge, visual localization benchmarks).

Other questionls:

(1) You claim the second contribution improves generalization — how do you show that?

(2) What happens if you add the same depth-assisted regularization to the detection branch? Please try it or explain why not.

**Questions:**

Please address my concerns in weaknesses.

---

### Official Review · Reviewer_SVxp · 2025-11-01

**Soundness:** 3
**Presentation:** 3
**Contribution:** 2
**Rating:** 4
**Confidence:** 4

**Summary:**

The paper proposes DXFeat, a depth-aware extension of the lightweight feature framework XFeat, whose main goal is to improve the robustness of local feature detection and description under strong viewpoint / depth / focal changes by introducing an auxiliary relative depth estimation task during training. To make this depth-aware auxiliary task actually help the main feature task and to keep training stable after adding it, the authors further introduce (1) a multi-level feature fusion with learnable layer weights, and (2) a Huber-style loss for reliability supervision. With these additions, DXFeat keeps almost the same inference-time cost as XFeat, but achieves consistently better matching / relative pose results on benchmarks such as MegaDepth, ScanNet, and HPatches.

**Strengths:**

1. DXFeat achieves higher accuracy than the almost identical XFeat architecture on indoor/outdoor relative pose estimation, while keeping the inference speed unchanged.
2. Compared to XFeat, DXFeat keeps the inference-time network simple and does not introduce extra runtime complexity.
3. To make the extra relative-depth auxiliary task actually help local feature learning, the authors add two stabilizing modules — (i) multi-level fusion with learnable weights and (ii) a Huber-style loss for reliability supervision.
4. The paper provides detailed ablation studies showing that each of these components contributes to the final performance.

**Weaknesses:**

1. **Limited gains in sparse matching.** Although DXFeat *consistently* outperforms XFeat in the **semi-dense matching setup**, the improvements become much smaller in the **sparse matching** regime, and on some metrics of MegaDepth and HPatches DXFeat is even slightly below XFeat.
2. **No downstream visual localization results.** The paper does not report results on downstream tasks such as visual localization, which are the most direct consumers of viewpoint-robust local features. The lack of such an evaluation is a notable omission and makes it harder to judge real-world impact.
3. **Related work on 3D-/depth-informed matching is under-discussed.** The paper would be stronger if it explicitly positioned DXFeat against recent “match-in-3D” methods such as [1] DUSt3R, [2, 3] (Speedy) MASt3R, and also [4] rectified-features approaches. These works also leverage 3D structure (often via point maps) to improve correspondence quality, so discussing how DXFeat differs would help. Likewise, although LiftFeat is included in experiments, it is highly related and deserves a short discussion in Related Work rather than only appearing in tables.

[1] DUSt3R: Geometric 3D Vision Made Easy

[2] Grounding Image Matching in 3D with MASt3R

[3] Speedy MASt3R

[4] Single-Image Depth Prediction Makes Feature Matching Easier

**Questions:**

1. DXFeat improves over XFeat much more in semi-dense than in sparse matching. What do you think causes this?
2. For the learnable layer-wise fusion, do you use just one global scalar per layer (shared for all images)? If so, what are the final learned weights/ratios?
3. If you keep the depth branch for analysis, what depth quality do you get on ScanNet and HPatches? Could poor/irrelevant depth on HPatches explain why DXFeat is slightly worse than XFeat on some HPatches metrics?

---

### Meta-Review · Area_Chair_YD8e · 2026-01-06

**Summary:**

Mainly concerns are lack of comparisons with other depth or 3D informed matching methods. 6Mbb raised the point that work is empirical, even ablation lacked  reasoning and results have been summarized without interpretation.
After careful analysis, my view is SVxp and puJS might have raised score by one point (currently they are at 4). However, QZLU and 6Mbb might not have.
Overall, requirement of experimentation would have been too much to answer all the concerns. This also indicates that paper needs careful revision.
Authors are encouraged to take inconsideration all the comments especially of 6Mbb to update their draft.

**Reviewer Concerns:**

None were addressed.

**Reviewer Scores:**

SVxp (4)  --- Might be
QZLU (2) --- No
6Mbb (4) --- No
puJS (4) --- Might be (but has very low confidence so increase in rank might not have helped).

---

### Decision · Program_Chairs · 2026-01-26

Reject